# Mistletoe Extracts Inhibit Progressive Growth of Prostate Cancer Cells

**DOI:** 10.3390/cells14191535

**Published:** 2025-09-30

**Authors:** Sascha D. Markowitsch, Larissa Albrecht, Moritz Meiborg, Jochen Rutz, Anita Thomas, Felix K.-H. Chun, Axel Haferkamp, Eva Juengel, Roman A. Blaheta

**Affiliations:** 1Department of Urology and Pediatric Urology, University Medical Center Mainz, 55131 Mainz, Germany; sascha.markowitsch@unimedizin-mainz.de (S.D.M.); lalbrech@students.uni-mainz.de (L.A.); jochen.rutz@unimedizin-mainz.de (J.R.); anita.thomas@unimedizin-mainz.de (A.T.); axel.haferkamp@unimedizin-mainz.de (A.H.); eva.juengel@unimedizin-mainz.de (E.J.); 2Department of Urology, Goethe-University, 60590 Frankfurt am Main, Germany; moritz.meiborg@web.de (M.M.); felix.chun@kgu.de (F.K.-H.C.)

**Keywords:** prostate cancer, mistletoe, growth, proliferation, cell cycling

## Abstract

Although multimodal therapeutic management has significantly improved outcome in prostate cancer (PCa) patients, treatment options for castrate-resistant disease remain challenging. Plant-derived mistletoe extracts have supported cancer patients and are, therefore, widely used as complementary medicine. However, mechanisms behind possible mistletoe benefits to PCa patients remain to be explored. The present study was designed to evaluate the effect of mistletoe extracts from four different host trees (*Tiliae*, *Populi*, *Salicis*, and *Crataegi*) on the growth and proliferation of PCa cell lines in vitro. PC3, DU145, and LNCaP cells were used to evaluate tumor cell growth (MTT assay) and proliferation (BrdU incorporation assay). Clonogenicity, apoptosis, cell cycle, and cell-cycle-regulating proteins (cyclin-dependent kinases (CDKs) and cyclins) were investigated, as was CD44 standard and splice variant expression and integrin α and β receptors. SiRNA knockdown studies were employed to investigate the functional relevance of integrins. All mistletoe extracts significantly inhibited cell growth in a dose-dependent manner and cell proliferation and clonogenicity were suppressed. *Populi* and *Salicis* induced cell-cycle arrest in the G2/M phase and increased apoptosis. Both extracts down-regulated CDK1 and cyclin A and altered CD44 expression. Integrins α5 in all cell lines and α6 in DU145 and LNCaP were particularly diminished. Knocking down α5 and α6 induced cell growth inhibition in DU145. Mistletoe extracts block the growth and proliferation of PCa cells in vitro and therefore qualify for use in future animal studies to evaluate mistletoe as an adjunct to standard PCa treatment.

## 1. Introduction

Prostate cancer (PCa) is the second most common cancer in men worldwide [1,2]. The risk of getting PCa increases with age, with the highest incidence in men over 65. Treatment options depend on the cancer stage, the aggressiveness of the tumor, state of health, and symptom development. Radical prostatectomy is the treatment of choice to remove the tumor with curative intent in localized or locally advanced PCa. However, since localized PCa usually progresses slowly, active surveillance is recommended in patients fulfilling low-risk criteria [3,4]. Irrespective of the kind of treatment, PCa-specific mortality has been proven to be similar in low-risk patients [5].

In contrast to the success of early stage PCa treatment, managing patients with advanced PCa remains challenging. The 5-year relative survival rate for men with local or regional PCa is nearly 100% while that for patients with metastasized PCa (mPCa) is estimated at 30% [6]. Several anticancer drugs have been approved for mPCa therapy during the last decade, leading to survival prolongation and improved quality of life [7]. Treatment options for hormone-sensitive mPCa include androgen deprivation therapy (ADT) combined with androgen receptor signaling inhibitors (ARSI) and docetaxel. To manage castration-resistant mPCa poly ADP ribose polymerase (PARP) inhibitors, as well as prostate-specific membrane antigen (PSMA)-targeted radioligand therapy are employed [8]. Notwithstanding these treatment advances, mPCa still remains incurable, and the search for new therapeutic approaches is ongoing.

Failure of or dissatisfaction with conventional cancer therapy drives many cancer patients to ask for complementary and alternative medicinal (CAM) therapeutic approaches, particularly incorporating plant derived isolated bioactive compounds and/or extracts. The proportion of patients applying herbal preparations has increased significantly during the last decade, reaching a mean value of about 50% worldwide [9,10]. The main motivations for CAM use are to improve the therapeutic efficacy of conventional cancer care, to improve quality of life [11,12], and to boost the immune system [13].

Among the plethora of available herbal compounds, extracts of European mistletoe (*Viscum album*) are the most popular, with a median of 26.7% (range: 7.3–46.3%) of cancer patients using them in German-speaking countries [14]. Mistletoe is most commonly used by patients with breast cancer [14]. Clinical trials with mistletoe in patients with pancreatic [15], lung [16], or colorectal cancer [17] have currently been initiated.

Despite mistletoe being increasingly applied in cancer treatment, outcome of studies with mistletoe are not always convincing, making its clinical relevance still controversial. Several investigations document improvement in quality of life and activation of the immune system [18,19]. Disease stabilization and improved survival have frequently been reported as well. However, due to weak study design, the validity of data from these studies often does not hold up under careful scrutiny [20]. Data regarding mistletoe’s molecular mode of action have not been provided in detail.

Nonetheless, preclinical studies carried out on breast cancer cells point to the induction of apoptosis and suppression of proliferation with mistletoe preparations [21,22]. Similar effects have been observed with further solid tumor types including liver [23], gastric [24], and lung cancer cells [25]. Mistletoe has been proven to target the mitogen-activated protein kinase (MAPK) and the PI3K/Akt pathway related to growth and survival of the cancer cells [26]. Furthermore, mistletoe extracts inhibit bladder cancer cell growth and proliferation, presumably by acting on the cyclin-CDK axis and the expression level of membrane receptors associated with tumor cell differentiation [27]. Whether mistletoe can be used to treat PCa remains to be explored. In fact, although studies have been carried out with different tumor entities [17,19], investigation with regard to PCa is sparse. This scarcity is not justified since mistletoe is applied by nearly 10% of PCa patients (related to CAM users in Austria) [28]. Büssing et al. assumed that mistletoe extracts may provide benefit in PCa patients when given on an individual basis [29]. Evidence has shown that mistletoe may cause an additive inhibitory effect on PCa cells when combined with chemotherapy [30].

Two phase I studies have meanwhile been initiated, documenting safety of mistletoe application in PCa patients [31] along with stable disease and tumor shrinkage in some heavily pretreated patients [17]. However, the authors pointed to the “preliminary” nature of their data and recommended further investigation [17]. Therefore, we initiated a comparative in vitro study with a panel of PCa cell lines to explore the growth and proliferation blocking potential of mistletoe preparations derived from four different host trees, *Tiliae*, *Populi*, *Salicis*, and *Crataegi*.

Besides assessing growth and proliferation, cell-cycle progression and the expression of proteins involved in cell-cycle regulation (cyclin-dependent kinases, cyclins) were analyzed in mistletoe treated versus non-treated PCa cells. This was carried out by investigating integrin α and β subtypes that have been shown to regulate PCa growth, progression, and metastatic formation [32]. Since the CD44s and CD44v isoforms are also involved in PCa proliferation, invasion, and metastasis [33], they were evaluated as well.

## 2. Materials and Methods

### 2.1. Cell Culture

PCa cell lines DU145 (RRID: CVCL_0105), PC3 (RRID: CVCL_0035) and LNCaP (RRID: CVCL_0395) were purchased from DSMZ (Braunschweig, Germany). Cells were grown and sub-cultured in Iscove Basal Medium (Biochrom GmbH, Berlin, Germany), supplemented with 10% fetal calf serum (FCS), 2% HEPES (2-(4-(2-Hydroxyethyl)-1-piperazine)-ethane sulfonic acid) buffer, 1% Glutamax (all Gibco/Invitrogen, Karlsruhe, Germany), and 1% penicillin/streptomycin (both Sigma-Aldrich, München, Germany) at 37 °C in a humidified 5% CO_2_ incubator.

### 2.2. Mistletoe Extracts

The mistletoe extracts Iscucin^®^ Tiliae, Populi, Salicis and Crataegi, host trees limetree (*Tiliae*), poplar (*Populi*), willow (*Salicis*) and hawthorn (*Crataegi*), were provided by Wala Heilmittel GmbH at a concentration termed H (1:20 from mother tincture) in an isotonic solution. The concentrations of mistletoe lectin and viscotoxin were as follows: lectin–Tiliae: 338.54 ng/mL, lectin–Populi: 184.33 ng/mL, lectin–Salicis: 246.46 ng/mL, lectin–Crataegi: 211.49 ng/mL, viscotoxin–Tiliae: 1.046 µg/mL, viscotoxin–Populi: 878 µg/mL, viscotoxin–Salicis: 980 µg/mL, viscotoxin–Crataegi: 1.109 µg/mL. The extracts, prepared according to an established protocol [34,35], were further diluted in phosphate-buffered saline (PBS) or cell culture medium for the experimental series. Isotonic solvent or cell culture medium alone was used as control. The employed dilutions were as follows, with (E) and (F) designating the dilutions used to evaluate proliferation (after 48 h incubation), or (E) to evaluate cell cycling (after 24 h incubation), apoptosis (after 48 h incubation), and integrin/CD44 expression (after 24 h incubation). To guarantee identical concentrations with the same extract composition, all dilutions were prepared from the same mistletoe batch. The following dilutions and intermediate dilutions were selected for cell growth and clonogenic growth:
Iscucin^®^ TiliaeIscucin^®^ PopuliIscucin^®^ SalicisIscucin^®^ Crataegi1:8 × 10^6^1:8 × 10^5^1:8 × 10^5^1:8 × 10^5^1:1.6 × 10^5^ (E)1:1.6 × 10^5^ (E)1:1.6 × 10^5^ (E)1:1.6 × 10^5^ (E)1:8 × 10^4^1:8 × 10^4^1:8 × 10^4^1:8 × 10^4^1:8 × 10^3^ (F)1:8 × 10^3^ (F)1:8 × 10^3^ (F)1:8 × 10^3^ (F)

### 2.3. Tumor Cell Growth

Cell growth (viability) was evaluated after 24, 48, and 72 h using the MTT (3-(4,5-dimethylthiazol-2-yl)-2,5-diphenyltetrazolium bromide) assay, following the manufacturer’s instructions (Roche Diagnostics, Penzberg, Germany). For each mistletoe extract treatment and time point, triplicates containing 5 × 10^3^ cells were pipetted into 96-well plates. Cells treated with medium served as controls. Absorbance at 550 nm was assessed with a microplate reader (Tecan Infinite M200, Männedorf, Switzerland). A standard range of cell numbers (from 2.5 × 10^3^ to 1.6 × 10^5^ cells per well) was used to convert absorbance into a cell count. After subtracting background absorbance (cell culture medium alone), results were expressed as mean cell number in percent. To evaluate dose–response kinetics, the mean cell number after 24 h incubation was set to 100%.

### 2.4. Tumor Cell Proliferation

To evaluate the proliferative activity of the tumor cells, BrdU (5-bromo-2′-deoxyuridine) incorporation was detected with the BrdU cell proliferation enzyme-linked immunosorbent assay (ELISA) kit (Calbiochem/Merck Biosciences, Darmstadt, Germany), according to the manufacturer’s instructions. Tumor cells (5 × 10^3^) were added to 96-well plates and then incubated with BrdU labeling solution for 24 h. Absorbance was measured at 450 nm using a microplate reader (Tecan Infinite M200, Männedorf, Switzerland). Values were expressed as percentages compared to untreated controls (set to 100%).

### 2.5. Clonogenic Growth

The clonogenic assay evaluates the ability of single cells to proliferate and form colonies, even from a single cell. LNCaP cells were not included in the clonogenic assay due to their inherently low basal cloning capacity. 500 cells/well were seeded onto 6-well plates for 10 days. The processes for stopping, fixing, staining, and analyzing the clones have been described previously [36]. A colony was defined as incorporating at least 50 cells. Untreated controls were set to 100%.

### 2.6. Cell-Cycle Analysis

Cell-cycle analysis performed using CycleTest™ Plus DNA Reagent Kit (BD Biosciences, Heidelberg, Germany) according to the manufacturer’s instructions. Tumor cells were then subjected to flow cytometry using FACSCalibur (BD Biosciences, Heidelberg, Germany). For each sample, 1 × 10^4^ events were observed. Data acquisition and calculation of cell-cycle distribution was carried out using CellQuest (BD Biosciences, Heidelberg, Germany). The percentage of gated cells in the G1, G2/M, or S phase was calculated using ModFit software (version 3.3; BD Biosciences, Heidelberg, Germany).

### 2.7. Apoptosis

PCa cells were removed with Accutase^®^ (PAA Laboratories GmbH, Pasching, Austria) and apoptosis/necrosis was determined. The Annexin V-FITC Apoptosis Detection Kit (BD Pharmingen, Heidelberg, Germany) was used to study apoptotic/necrotic events according to the manufacturer’s instructions. Samples were analyzed using a FACScalibur (BD Biosciences, Heidelberg, Germany); 1 × 10^4^ cells per scan. The percentage of apoptotic (early and late), necrotic, and viable cells was calculated using CellQuest software (version 5.1; BD Biosciences, Heidelberg, Germany).

### 2.8. CD44 Expression

PCa cells were detached from the culture flasks (Accutase^®^; PAA Laboratories GmbH, Pasching, Austria), washed, and blocked using PBS with 0.5% bovine serum albumin (BSA). They were then incubated for 1 h at 4 °C with 20 µL allophycocyanin (APC)-coupled monoclonal antibodies. To investigate CD44 standard (CD44s) or CD44 variants (CD44v) expression, all antibodies, anti-CD44s (clone SFF-2, RRID:AB_10597135), anti-CD44v3 (clone VFF-327v3, RRID:AB_10597004), anti-CD44v4 (clone VFF-11, RRID:AB_10596817), anti-CD44v5 (clone VFF-8, RRID:AB_10598046), anti-CD44v6 (clone VFF-7, RRID:AB_10596818), and anti-CD44v7 (clone VFF-9, RRID:AB_10597005; all: eBioscience, ThermoFisher), were conjugated to APC using the Lightning-Link APC Conjugation Kit (eBioscience, ThermoFisher, Darmstadt, Germany) and left to attach to the PCa cells for 60 min at 4 °C. Mouse IgG1, K (clone P3.6.2.8.1, RRID: AB_2865977; ThermoFisher, Dreieich, Germany), coupled to APC, acted as the isotype control. Analysis was performed by a FACSCalibur (BD Biosciences, Heidelberg, Germany); FL4-H (CD44s/CD44v) (log) channel histogram analysis; 1 × 10^4^ cells per scan). Fluorescence values were expressed as mean fluorescence units (MFU).

### 2.9. Integrin Surface Expression

PCa cells were detached using Accutase^®^ (PAA Laboratories GmbH, Pasching, Austria), washed, and blocked using PBS with 0.5% BSA. They were then incubated for 1 h at 4 °C with 20 µL of phycoerythrin (PE)-conjugated monoclonal antibodies directed against the following integrin subtypes: anti-α1 (IgG1; clone SR84, RRID: AB_397288), anti-α2 (IgG2a; clone 12F1-H6, RRID: AB_396022), anti-α3 (IgG1; clone C3II.1, RRID: AB_396301), anti-α4 (IgG1, clone 9F10, RRID: AB_395893), anti-α5 (IgG1; clone IIA1, RRID: AB_395984), anti-α6 (IgG2a; clone GoH3, RRID: AB_775720), anti-β1 (IgG1; clone MAR4, RRID: AB_395836), anti-β3 (IgG1; clone VI-PL2, RRID: AB_396095) or anti-β4 (IgG2a; clone 439-9B, RRID: AB_396063; all BD Biosciences, Heidelberg, Germany). The integrin expression of the tumor cells was then evaluated using FACSCalibur (BD Biosciences; FL2-H (log) channel histogram analysis; 1 × 10^4^ cells per scan) and expressed as mean fluorescence units. Mouse IgG1-PE (clone MOPC-21, RRID: AB_397218), mouse IgG2a-PE (clone G155-178, RRID: AB_396517) or rat IgG2b-PE (clone R35-38, RRID: AB_396171) (all BD Biosciences, Heidelberg, Germany) were used as isotype controls.

### 2.10. Western Blot Analysis

Western blot was used to evaluate the expression and activity of cell cycle and death-regulating proteins. Tumor cell lysates (50 µg) were applied to 10% or 12% polyacrylamide gel and separated for 10 min at 80 V and 1 h at 120 V. The protein was then transferred to nitrocellulose membranes (1 h, 100 V). After blocking with 10% non-fat dry milk for 1 h, the membranes were incubated overnight with the following primary antibodies directed against cell-cycle proteins: cyclin A (Mouse IgG1, clone 25, RRID: AB_398797, dilution 1:500), cyclin B (Mouse IgG1, clone 18, RRID: AB_397616, dilution 1:1000), cyclin D3 (Mouse IgG2b, clone 1, RRID: AB_397675, dilution 1:1000), cyclin E (Mouse IgG2b, clone HE67, RRID: AB_399161, dilution 1:1000), CDK1 (Mouse IgG1, clone 1, RRID: AB_397454, dilution 1:2500), CDK2 (Mouse IgG2a, clone 55, RRID: AB_397546, dilution 1:2500; all BD Biosciences, Heidelberg, Germany).

Horseradish peroxidase (HRP)-conjugated goat-anti-mouse or goat-anti-rat-IgG (Cell Signaling Technology, Cambridge, UK; dilution 1:3000) served as the secondary antibodies. To visualize proteins, membranes were incubated with an enhanced chemiluminescence (ECL) detection reagent (ECLTM, Amersham/GE Healthcare, Munich, Germany). After incubation, the membranes were analyzed using the Fusion FX7 system (Peqlab, Erlangen, Germany). β-actin (clone AC-15, RRID: AB_476744, 1:10,000; Sigma, Taufenkirchen, Germany) was used as an internal control. Pixel density analysis of the protein bands was performed with Gimp 2.8.20 software (www.gimp.org, assessed on 3 March 2025) before calculating the ratio of protein intensity/β-actin intensity.

### 2.11. Blocking Studies

DU145 cells were incubated for 60 min with 10 μg/mL function-blocking anti-integrin α5 (Mouse, clone P1D6, RRID: AB_94455) or α6 (Rat, clone NKI-GoH3, RRID: AB_2128317; all from Merck Millipore, Darmstadt, Germany) and then subjected to the MTT assay (see 2.3 Tumor Cell Growth). Transfection with small interfering RNA (siRNA) was carried out directed against CDK1 (gene ID: 983, target sequence: AAGGGGTTCCTAGTACTGCAA) or CDK2 (gene ID: 1017, target sequence: AGGTGGTGGCGCTTAAGAAAA; all: Qiagen, Hilden, Germany). 3 × 10^5^ cells were incubated for 24 h with a transfection solution containing siRNA and the transfection reagent (HiPerFect Transfection Reagent; Qiagen) at a ratio of 1:6. Untreated cells and cells treated with control siRNA (AllStars Negative Control siRNA; Qiagen) served as controls. The protein expression level was evaluated by Western blotting. Tumor cell growth was evaluated by the MTT assay (2.3.).

### 2.12. Statistics

The mean +/− SD was calculated. Graphs were prepared using SigmaPlot 11 (upgrade version 14.5; SYSTAT Software, San Jose, CA, USA). To exclude coincidence, all experiments were repeated three to five times. Here, statistical significance was calculated with GraphPad Prism 7.0 (GraphPad Software Inc., San Diego, CA, USA). Statistical significance was evaluated with two-way ANOVA, one-way ANOVA or “Student’s *t*-test”. *p* < 0.05 was considered significant.

## 3. Results

### 3.1. Extracts from Mistletoe Inhibit Tumor Cell Growth and Proliferation of PCa Cells

Mistletoe extracts from the host trees *Tiliae*, *Populi*, *Salicis*, and *Crataegi* inhibited the growth of all three PCa cell lines in a time- and dose-dependent manner (Figure 1(A1–C4)). The degree of inhibition differed according to the extract origin and the cell line treated. *Crataegi*, diluted at 1:1.6 × 10^5^ (dilution E), suppressed cell counts in all three cell lines, but this dilution did not alter the growth of PC3 cells when derived from *Populi* or *Salicis*. The highest extract concentration, diluted 1:8 × 10^3^ (dilution F), completely blocked tumor growth in all experimental settings. The LNCaP cell line was particularly sensitive to treatment with the mistletoe extracts (Figure 1(C1–C4)).

No differences in cell growth activity were seen in the presence of isotonic solvent or cell culture medium alone (Figure 2A–D). Control values are, therefore, derived from cells cultured in cell culture medium, not in solvent.

Dilutions E (1:1.6 × 10^5^) and F (1:8 × 10^3^) were employed for further studies on cell proliferation and clonogenicity. Cell proliferation was significantly inhibited by all mistletoe extracts applied (Figure 2A–D). Dilution F was superior to dilution E, except for *Tiliae*, where both E and F were similarly effective in blocking the proliferation of LNCaP cells. DU145 cells were most sensitive to drug treatment. Particularly, *Populi* completely stopped BrdU uptake in DU145 when added at dilution F. A very strong response was induced when DU145 cells were exposed to *Salicis*. Therefore, ongoing studies concentrated on *Populi* and *Salicis*.

Subsequent clonogenicity evaluation was conducted using mistletoe extracts harvested from *Populi* and *Salicis*. LNCaP cells were excluded from this study since they did not form clones. Both *Populi* and *Salicis* significantly inhibited clonogenic growth of DU145 and PC3 cells to a similar extent, already at dilution E. Considerable loss of tumor clones was already induced by the extracts when applied at a dilution of 1:8 × 10^4^. No clones were seen at all in the presence of dilution F (Figure 3A–D). No differences in clonogenicity were seen in the presence of isotonic solvent or cell culture medium alone. Control values shown in the figure are from cells cultured in cell culture medium. Since clonogenic growth of DU145 and PC3 was similarly reduced, cell cycling and apoptosis were subsequently evaluated only in DU145 cells.

### 3.2. Mistletoe Extracts Inhibit the Cell Cycle of DU145 Cells

Cell cycling of DU145 cells was evaluated using mistletoe extracts from *Salicis* and *Populi*. Both extracts enhanced the number of cells in the G2/M phase. This enhancement was paralleled by a loss of cells in G0/G1 (Figure 4A). The G2/M accumulation was associated with alterations of cell-cycle regulatory proteins (Figure 4B,C, see also Appendix A).

CDK1, cyclin A, and cyclin D3 expression was significantly reduced, independent of the extract used. Cyclin B was significantly suppressed in the presence of *Salicis*, but not in the presence of *Populi* (Figure 4B,C).

### 3.3. Mistletoe Extracts Induce Apoptosis in DU145 Cells

Apoptosis was evaluated in the DU145 cell line at dilution E. Treatment with *Populi* and *Salicis* significantly increased the number of cells in early and late apoptosis, compared to cells treated with cell culture medium or solvent alone (Figure 5A–D). The number of necrotic cells remained unchanged.

### 3.4. Mistletoe Extracts Modulate CD44 and Integrin Expression in PCa Cells

CD44 std was strongly expressed on untreated DU145 and PC3 but only moderately expressed on LNCaP cells (Figure 6). CD44v3, v4, and v6 were distinctly expressed on all cell lines (LNCaP > DU145/PC3). CD44v5 and v7 were only moderately expressed.

Integrin α and β expression profiles of DU145, PC3, and LNCaP cells are shown in Figure 7. Integrin α2, α3, α5, α6, and β1 were detectable in all three cell lines with lower expression levels of α2 and α3 in LNCaP, compared to DU145 and PC3. The integrin subtypes α1 and β4 were only expressed on DU145 and PC3, while β3 was particularly strongly expressed on DU145. Integrin α4 was not expressed on any of the three cell lines.

*Populi* and *Salicis* altered the CD44 and integrin expression pattern of PCa cells (Figure 8A–F). The response was not homogeneous but rather varied among the cell lines. In DU145 cells, CD44v4 and v5 increased in the presence of *Salicis*, whereas v7 was diminished by both *Populi* and *Salicis* (Figure 8A). In contrast, *Salicis* reduced CD44v3, v5, v6, and v7 expression in PC3 cells (Figure 8B). However, CD44v5 was elevated in this cell line by *Populi*. CD44v3, v5, and v6 were down-regulated by both *Populi* and *Salicis*, whereas only *Populi* reduced CD44v4 and v7 in LNCaP cells (Figure 8C).

Integrins α2, α3, and β1 were up-regulated in DU145 cells by *Salicis* (Figure 8D). However, *Populi* and *Salicis* reduced α5 and α6 to the same extent. A similar effect was seen in PC3 cells whereby integrins α3 and β1 were enhanced by *Salicis* as well. α5 was lowered by both *Populi* and *Salicis* (Figure 8E). In accord with the action on DU145 cells, *Populi* and *Salicis* induced an α5 and α6 loss in LNCaP cells (Figure 8F). Integrins α2 and β1 were also diminished by both compounds, and *Populi* suppressed integrin α3 in LNCaP cells (Figure 8F).

Since *Populi* and *Salicis* diminished integrin α5 in all cell lines, and integrin α6 was down-regulated on DU145 and LNCaP cells, the physiologic relevance of these integrins was investigated by blocking studies. Blocking integrin α5 significantly inhibited growth of DU145 cells by approximately 50%. An even stronger growth inhibition was seen following integrin α6 blockade (Figure 9A). Knockdown of CDK1 and 2 also significantly reduced the growth of DU145 cells (Figure 9B). At the protein level, knockdown was shown to reduce CDK1 and 2 expressions (Figure 9C, see also Appendix A).

## 4. Discussion

Mistletoe (*Viscum album*) extracts prepared from the host trees *Tiliae*, *Populi*, *Salicis*, and *Crataegi* inhibited cell growth and proliferation of DU145, PC3, and LNCaP cells in a time- and dose-dependent manner. Clone formation in PC3 and DU145 cells was suppressed after exposure to *Populi* and *Salicis* in a dilution range of 1:8 × 10^3^–1:8 × 10^5^. Thus, PCa adds to the cancer types that have growth and proliferation inhibited by exposure to mistletoe. When applied in a similar dilution range these mistletoe extracts have also been shown to suppress growth and proliferation in a panel of bladder carcinoma cell lines [27]. Another study has demonstrated growth and proliferation-blocking effects of *Viscum album* extracts on bladder cancer cells at concentrations of 10, 100, and 1000 µg/mL [38], corresponding to a dilution of the mother tincture of 1:1 × 10^3^–1:1 × 10^5^. In breast cancer cells, mistletoe extracts have been shown to suppress growth and colony formation within a range of 62.5–1000 µg/mL (1:1 × 10^3^–1:1.6 × 10^4^ dilution) [39]. The potential of mistletoe (*Salicis*, 500 µg/mL, corresponding to a 1:2 × 10^3^ dilution of the mother tincture) as an antitumor agent has also been documented in renal cell carcinoma cell lines [35]. In vitro inhibitory effects of mistletoe are paralleled in vivo, where extracts have been shown to shrink hepatocellular [23] and mammary carcinomas in mice [40].

In the present investigation, the tumor cell response was not homogeneous but rather depended on both the mistletoe host tree and the specific cell line. *Viscum album* extracts derived from *Crataegi* reduced the cell count from all tumor cell lines equally well, even at a dilution of 1:8 × 10^5^. In contrast, *Tiliae* extracts most prominently exerted effects on DU145 cells where a dilution of 1:8 × 10^6^ lowered the tumor cell number by nearly 50%. The same dilution had no effect on PC3 cells. Cell line specific differences in the presence of *Tiliae* were also observed when examining proliferation. The reason for this divergent sensitivity can only be speculated upon. DU145 and PC3 both represent androgen-resistant tumor cells. However, PC3 and DU145 are derived from different metastatic sites, PC3 from a bone metastasis (grade IV), DU145 from a brain metastasis (grade II) of PCa origin, with DU145 being less metastatic, compared to PC3 [41].

Hypothetically, the antitumor activity of particular mistletoe extracts (*Tiliae* in the present setting) may, at least partially, depend on the tumor cell differentiation status. Indeed, in a recent investigation with several bladder cancer cell lines, TCCSup cells (derived from an undifferentiated grade 4 tumor) were found to be less sensitive to *Tiliae* than UMUC3 and RT112 cells (both corresponding to a grade 2 tumor) [27]. Accordingly, cell growth of UMUC3 cells was blocked more potently by *Tiliae*, compared to TCCSup in another investigation [42]. Transferred to a clinical situation, PCa patients may particularly profit from mistletoe when integrative treatment is initiated immediately after diagnosis, i.e., when the tumor has not spread to distant organs.

A correlation between mistletoe efficacy and initial tumor differentiation status has also been observed in breast cancer cells. In contrast to our results with PCa cells, the highly metastatic MDA-MB-231 cell line was more sensitive to mistletoe treatment than the less metastatic MCF-7 cell line [43]. Mistletoe from the host tree *Quercus* was used in the breast cancer investigation, indicating that mistletoe’s mode of action depends on the tumor entity and the host tree. Comparing tumor-specific differences has shown that lung cancer cells are highly sensitive to mistletoe while gastric cancer cells are resistant [25]. A systematic literature review with meta-regression for cancer entities has shown significant differences in sensitivity between lung cancer and the reference class breast cancer to the *Viscum album* plant extract Iscador [44,45].

Host tree and seasonal aspects have also been correlated with the cytotoxic potential of mistletoe extracts [46]. Extracts from a summer harvest exhibited a more potent cytotoxic effect than from winter harvests, and mother tinctures prepared from the subspecies *Viscum album album* exerted stronger antitumor activity, compared to tinctures from *Viscum album abietis* or *Viscum album austriacum* [46]. The reason for these differences remains unclear. Chemical analysis shows substantial differences in the composition of pharmacological compounds in mistletoe extracts prepared from the different host trees [47]. In particular, the content of mistletoe lectin and viscotoxin varies among the host tree species and between summer and winter harvesting [46,47]. Both compounds are probably related to the antitumor effect of mistletoe extracts. Viscotoxin and lectin concentrations also differed in the mistletoe extracts used in the present study. However, viscotoxin was highest in extracts derived from *Crataegi*, whereas lectin was highest in extracts derived from *Tiliae*, making it difficult to establish a relationship between these substances and tumor cell response. In this context, it has been postulated by others that the whole mistletoe content in concert, rather than single compounds, exerts effects on tumor cells. Analysis of the metabolome of 50 *Viscum album* mother tinctures notes a total of 188 metabolites, with 14 compounds thought to be responsible for cytotoxicity [46]. Therefore, the different responses of the PCa cells to the mistletoe extracts used here may not be exclusively related to the viscotoxin and lectin content but to the composition of the total phytocomplex [26].

Different protein function in the different PCa tumor lines may also account for the varying responses to mistletoe treatment. The cell lines used in the present investigation have different functional p53 activity. LNCaP expresses wild-type p53, DU145 carries p53 mutations, and PC3 has a deleted p53 gene. This is noteworthy, since the functional status of p53 has been correlated with treatment success [48,49]. The same could hold true for the tumor suppressor PTEN and PTEN-related Akt signaling. Both PC3 and LNCaP are PTEN-deficient with hyperactivated Akt, while DU145 has a high PTEN expression level and low Akt activity [50]. Mistletoe extracts from *Viscum album* have been shown to affect squamous cell carcinoma cell lines by altering Akt and PTEN [51]. The relevance of the PCa protein expression profile and treatment response requires further evaluation.

We attribute the growth and proliferation inhibition in DU145 cells observed after exposure to mistletoe extracts to cell-cycle arrest in G2/M. G2/M phase arrest induced by *Viscum album* extracts has also been shown in glioblastoma cells (host tree: *Quercus*) [52] and in breast carcinoma cells (host tree: *Malus domestica*) [53]. The Korean mistletoe *Viscum album coloratum*, a subspecies of European mistletoe, has also been shown to induce G2/M phase arrest in MDA-MB-231 breast cancer cells [54]. A G2/M phase arrest is, however, not universal in cancer cells. UMUC3 cells treated with mistletoe from *Populi* or *Salicis* were arrested at G0/G1 [27]. An increase in the G1 phase in hepatocellular carcinoma cells has also been seen after exposure to hot water extracts of *Viscum album var* [55]. Whether mistletoe exposure results in G2/M arrest in LNCaP or PC3 cells is unknown. There is a report that the mistletoe plants *Plicosepalus curviflorus* and *Plicosepalus acacia*, both from the family *Loranthaceae*, induced a PC3 cell-cycle block at the Pre-G1 and G1 phase [56]. Probably, the reaction of the cell cycle to mistletoe extracts in different cancer cells depends on both the host tree and the type of cancer cell line.

Differences were seen in regard to the modification of cell-cycle-regulating proteins by *Populi* and *Salicis*. CDK1 expression was moderately suppressed by *Populi* in DU145 cells, whereas this protein was completely lost following exposure to *Salicis*. *Salicis*, but not *Populi*, also reduced cyclin B expression. Similarly, in bladder cancer cells CDK2 was nearly lost in the presence of *Salicis* but reduced to a lesser extent by *Populi*. CDK1 was suppressed by *Salicis* only [27]. It was concluded that these differences might account for the different effects of *Salicis* and *Populi* on bladder cancer proliferation and apoptosis. The present investigation points to a stronger suppression of DU145 colony formation and induction of apoptosis by *Salicis*, compared to *Populi*. Since down-regulation of cyclin B and CDK1 has been closely associated with DU145 phase arrest in G2/M, along with enhanced cell apoptosis and clone formation of this cell line [57,58], the different influence of *Salicis* and *Populi* on the CDK1-cyclin B axis might contribute to the different antitumor potential of these compounds. We show the relevance of CDK1 and CDK2 to DU145 growth with siRNA knockdown studies where growth was inhibited when CDK1 and CDK2 were knocked down.

The role CD44 may play in PCa is ambiguous. CD44v3 and CD44v6 expressed on PC3 cells has been shown to correlate with migration [59]. Contradicting this, other investigators have shown down-regulation of CD44v6 to be related to high T classification, metastasis, and a high Gleason score [60]. There is also evidence that PC3 cells, transfected with CD44v3-v10, lose adhesiveness and invasiveness, compared to wild-type PC3 [61]. Moura et al. has presented evidence that PCa cases are characterized by an overexpression of CD44 variants which, however, is lost during cancer progression [62]. CD44 variants v4, v5, and v7 have been associated with reduced PCa growth and proliferation [63] but elevated PCa invasion in vitro [64]. Thus, the relevance of CD44v expression upon mistletoe exposure cannot satisfactorily be resolved. We found that the CD44 expression pattern in the different PCa cell lines was differently altered by *Populi* and *Salicis*. It should be noted that the basal CD44 expression pattern varies among the PCa cell lines investigated. The p53 and PTEN status also differed cell line specifically. This is noteworthy, since CD44-p53 cross-communication [65,66] as well as CD44-PTEN [67,68] or PTEN-p53 interactions [69] have been shown in tumor cells. Due to the different receptor and protein constellations in the PCa cell lines, different cell signaling mechanisms under mistletoe treatment could be expected. However, this is speculative and requires verification. It remains open whether the different CD44 responses can actually be associated with mistletoe’s inhibitory effect on PCa growth and proliferation that is similar among the cell lines.

Several integrin α and β subtypes were also altered non-homogeneously by *Populi* and *Salicis*, excepting for α5 and α6 which were diminished on DU145, PC3 (only α5 significantly), and LNCaP. Knockdown of α5 and α6 significantly reduced DU145 cell growth, indicating that cell growth blockade by the mistletoe extracts might (at least partially) be traced back to lowered α5 and α6 expression. Sufficient evidence has been provided that integrin α5 plays a decisive role in PCa progression. Preclinical PCa models point to α5 as a driver of tumor growth and metastasis [70]. Integrin α5 enrichment in bone-metastatic tumors has been documented in samples from PCa patients, underscoring the role of α5 in tumor colonization of the bone microenvironment [71].

The role of integrin α6 is less clear. Both oncogenic and tumor-suppressing functions have been ascribed to α6. Systematic reviews and meta-analyses have identified integrin α6 as a candidate biomarker for PCa risk [72]. In DU145 cells, the α6 expression level positively correlated with cellular adhesion and invasion capacity [73]. This contrasts with another study where disruption of hemidesmosomes, by depleting α6 and its binding partner β4, promoted cell migration of PCa cells. However, these investigators argued that disruption of hemidesmosomes with loss of the β4 integrin was responsible for facilitating cell migration [74]. In line with this explanation, integrin β4 reduction in concert with loss of hemidesmosomal adhesions correlated with higher PCa metastatic capacity, while integrin α6 reduction plus loss of hemidesmosomal adhesions forced migration only in PTEN-deficient cells [75].

Integrin α6 has been shown to activate the PI3K/Akt pathway when hemidesmosomes are intact [76]. A recent study on a panel of PCa cell lines documented cross-communication between integrins and Akt/mTOR signaling, closely associated with tumor cell growth capacity [37]. Based on the in vitro data in the present investigation, we propose that mistletoe extracts inhibit PCa growth through integrin α5 and α6 suppression. In-depth analyses are now in progress to explore whether integrins α5 and α6 are connected to the Akt/mTOR pathway.

## 5. Conclusions

Mistletoe extracts exert distinct growth- and proliferation-blocking effects on PCa cells in vitro. The molecular mode of action differed among the tumor cell lines. Differences were also observed depending on the mistletoe host tree. Consequently, PCa patients may not profit from mistletoe treatment equally well, and a particular extract may not evoke the same response in every patient and may depend on tumor staging. Further studies should be directed towards the aspect of inhomogeneous response to mistletoe. It is of particular interest to evaluate how the tumor cell response to the extracts might be associated with the tumor cell differentiation status. Since specific cell signaling pathways altered by the mistletoe extracts may also be influenced by conventional treatment, it is important to integrate mistletoe extracts into first-line treatment in preclinical experiments. Undesired protein cross-communication could reduce the tumor cell response to drug treatment. Since drug resistance remains the major problem in cancer therapy, it is now intended to investigate the influence of mistletoe extracts on drug-resistant PCa cell lines.

## Figures and Tables

**Figure 1 cells-14-01535-f001:**
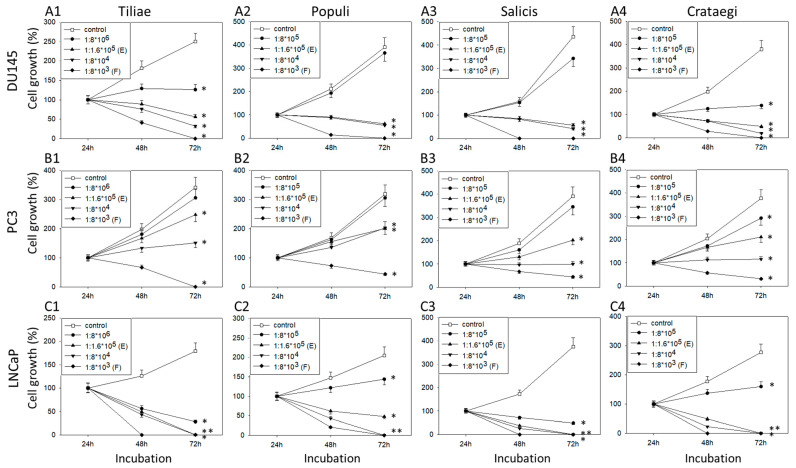
Growth of PCa cells after treatment with mistletoe extracts. Cell growth (viability) of DU145 (**A1**–**A4**), PC3 (**B1**–**B4**), and LNCaP (**C1**–**C4**) cells was examined after exposure to mistletoe extracts from *Tiliae* (**A1**,**B1**,**C1**) diluted 1:8 × 10^6^–1:8 × 10^3^, from *Populi* (**A2**,**B2**,**C2**) diluted 1:8 × 10^5^–1:8 × 10^3^, from *Salicis* (**A3**,**B3**,**C3**) diluted 1:8 × 10^5^–1:8 × 10^3^, and from *Crataegi* (**A4**,**B4**,**C4**) diluted 1:8 × 10^5^–1:8 × 10^3^ for 24, 48, and 72 h. Values are normalized to corresponding 24 h values = 100%. Error bars indicate standard deviation (SD). Significant difference to untreated control (cells exposed to culture medium alone): * *p* ≤ 0.05. *n* = 4.

**Figure 2 cells-14-01535-f002:**
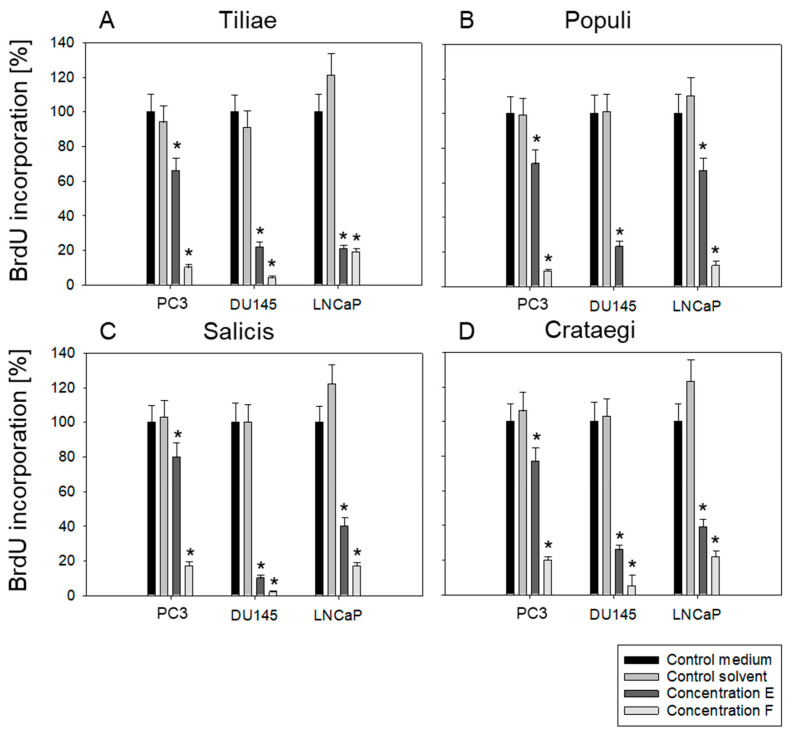
Proliferation of PCa cells after exposure to mistletoe extract. Cell proliferation (BrdU incorporation) after treatment with *Tiliae* (**A**), *Populi* (**B**), *Salicis* (**C**) and *Crataegi* (**D**) at dilutions E [1:1.6 × 10^5^] and F [1:8 × 10^3^] after 48 h incubation. Cells with medium instead of mistletoe extract served as controls and were set to 100%. Error bars indicate standard deviation (SD). Significant difference to untreated control: * *p* ≤ 0.05. *n* = 4.

**Figure 3 cells-14-01535-f003:**
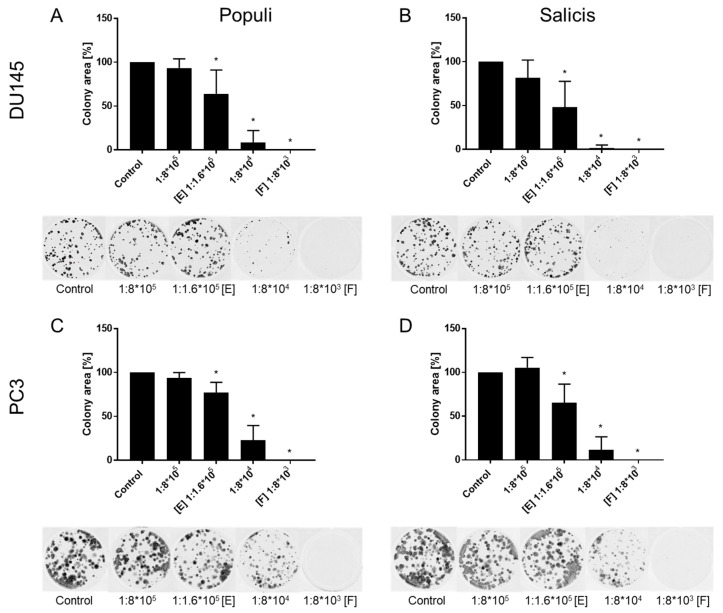
Clonogenic growth of PCa cells after treatment with mistletoe extracts. Cells exposed to *Populi* or *Salicis* [1:8 × 10^3^–1:8 × 10^5^ dilution] (**A**–**D**) for 10 days. Untreated cells served as controls and were set to 100%. In each case, a bar chart on the top, and a respective representative depiction (**A**–**D**) is shown on the bottom. Error bars indicate standard deviation (SD). Significant difference to untreated control: * *p* ≤ 0.05. *n* = 5.

**Figure 4 cells-14-01535-f004:**
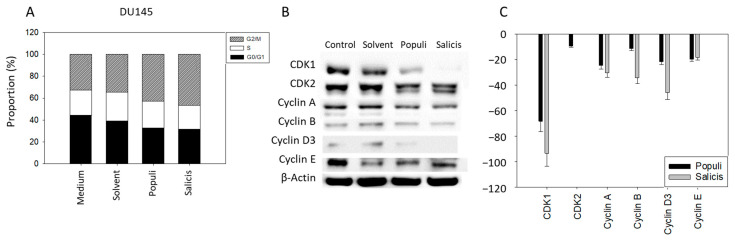
Cell-cycle analysis of DU145 cells after treatment with mistletoe extracts. Distribution of cell-cycle phases of untreated, solvent-treated, *Populi*-treated [dilution E], and *Salicis*-treated [dilution E] DU145 cells (**A**). Expression of cell-cycle regulatory proteins CDK1, CDK2, cyclin A, cyclin B, cyclin D3, and cyclin E of untreated, solvent-treated, *Populi*-treated [dilution E], and *Salicis*-treated [dilution E] DU145 cells after 24 h incubation (**B**). β-Actin served as the housekeeping protein. Expression of cell-cycle regulatory proteins CDK1, CDK2, cyclin A, cyclin B, cyclin D3, and cyclin E of *Populi* [dilution E] treated and *Salicis* [dilution E] treated DU145 cells as bar graph normalized to untreated control (=0) (**C**). Error bars indicate standard deviation (SD). Significant difference to untreated control: *n* = 4.

**Figure 5 cells-14-01535-f005:**
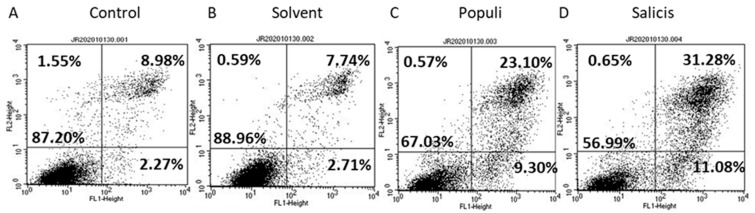
Apoptosis induction after treatment with mistletoe extracts in DU145 cells. Apoptosis of untreated (**A**), solvent-treated (**B**), *Populi*-treated [dilution E] (**C**), and *Salicis*-treated [dilution E] (**D**) DU145 cells after 48 h incubation. FL1 and FL2 negative show vital cells (bottom left square). FL1 positive and FL2 negative show early apoptotic cells (bottom right square). FL1 positive and FL2 positive show late apoptotic cells (top right square). FL1 negative and FL2 positive show necrotic cells (top left square). Figure shows one representative from *n* = 4. Mean values are given as percentage, SD_intraassay_ < 30%.

**Figure 6 cells-14-01535-f006:**
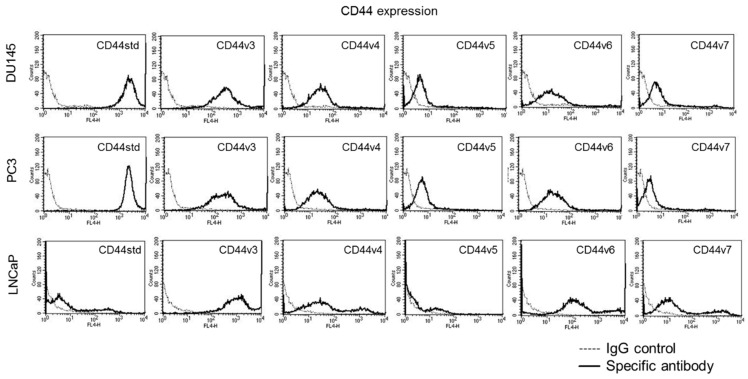
Basal CD44 surface expression in PCa cells. Basal CD44 expression of DU145, PC3, and LNCaP cells. Expression of CD44 standard (CD44std) and CD44 variant subtypes v3–v7 compared with the corresponding IgG isotype control. One representative from *n* = 4 is shown.

**Figure 7 cells-14-01535-f007:**
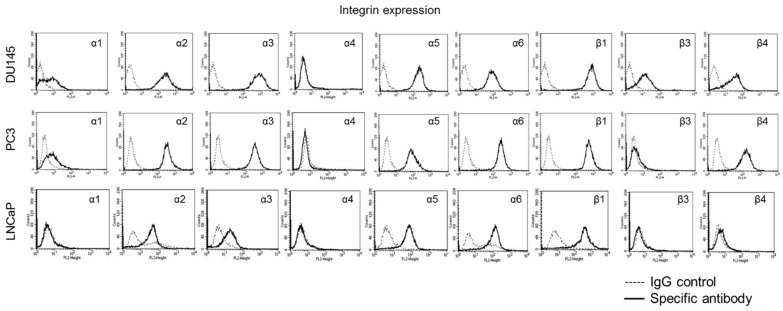
Basal integrin expression in PCa cells. Basal α and β integrin surface expression of DU145, PC3, and LNCaP cells (The data on basal expression of DU145 and PC3 cells are from a previous study and have been included here for completeness [37]). Expression of integrin subtypes α1-6 and β1, 3, and 4 compared with the corresponding IgG isotype control. One representative from *n* = 4 is shown.

**Figure 8 cells-14-01535-f008:**
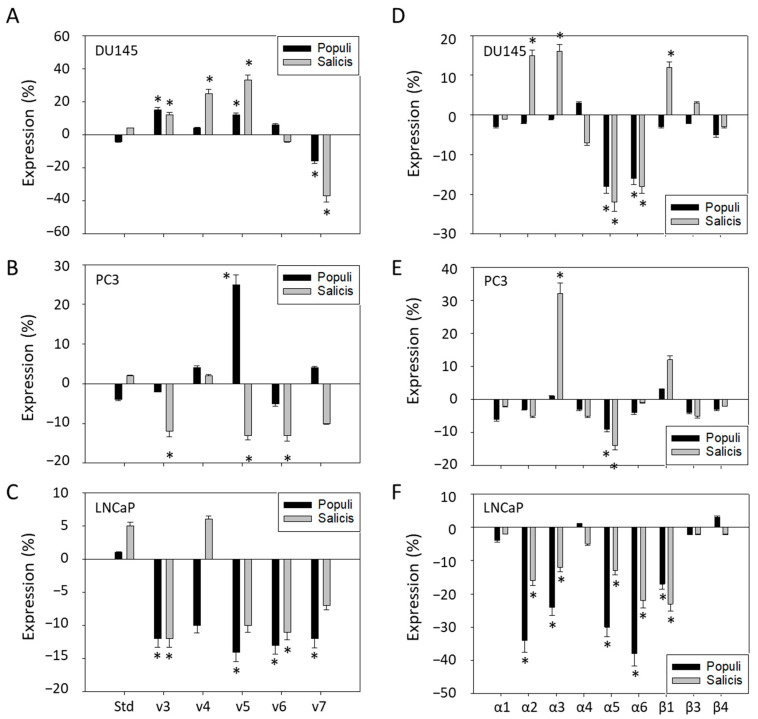
Integrin and CD44 expression after mistletoe extract treatment of PCa cells. Altered CD44 subtype expression (v3–v7) (**A**–**C**) and integrin subtype expression (α1–α6 and β1, β3, β4) (**D**–**F**) after 24 h exposure to *Populi* [dilution E] or *Salicis* [dilution E] mistletoe extracts, compared to the untreated control (=0) in DU145 (**A**,**D**), PC3 (**B**,**E**), and LNCaP (**C**,**F**) cells. Error bars indicate standard deviation (SD). Significant difference to untreated control: * *p* ≤ 0.05. *n* = 4.

**Figure 9 cells-14-01535-f009:**
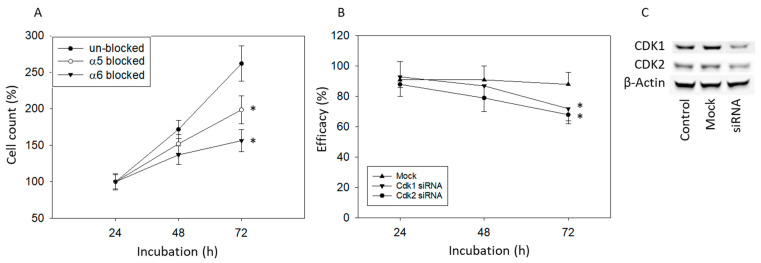
Influence of integrins α5 or α6 functional blocking and CDK1/CDK2 knockdown on DU145 cell growth. Growth of DU145 cells after functional blocking of integrins α5/6 (**A**) and siRNA blockade of CDK1/2 (**B**) at 24, 48, and 72 h. Control of the knockdown at the protein level in the Western blot (**C**). Untreated cells served as controls. Normalized to the corresponding 24 h value = 100%. Error bars indicate standard deviation (SD). Significant difference to untreated control: * *p* ≤ 0.05. *n* = 4.

## Data Availability

The original contributions presented in this study are included in the article/Appendix A. Further inquiries can be directed to the corresponding author(s).

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
