# Peer review of "Mistletoe Extracts Inhibit Progressive Growth of Prostate Cancer Cells"

_cells, 2025, doi:10.3390/cells14191535_

Round 1

Reviewer 1 Report

Comments and Suggestions for Authors

Markowitsch et al. analyzed extracts from mistletoes grown on four different host tree species (Tiliae, Populi, Salicis, Crataegi) as potential anti-cancer agents in their study. They investigated the effects of these different extracts on the behavior of three different prostate cancer (PCa) cell lines (DU145, PC3, LNCaP) at different cellular levels and also analyzed the effects on different CD44 variants and integrins in more detail. The study fits thematically well with the Special Issue “Natural Products and Their Derivatives Against Human Disease”.

In the introduction, the authors discuss the use of complementary and alternative medicinal (CAM) therapeutic approaches and point to initial clinical attempts and studies with mistletoe extracts. However, why they chose PCa as a tumor model should be better explained, as there seems to be no evidence of the efficacy of mistletoe preparations for this tumor entity so far. In addition, the clinical situation in which such preparations could be used in PCa should be considered more thoroughly.

If there are further reports than those already mentioned on which cellular processes mistletoe extracts may have an influence, these should be described in the introduction. There may also be more references on the role of CD44 and integrins in PCa already in the introduction. This would better justify the more detailed consideration of these and no other molecules and pathways.

The methods are adequately introduced and explained. However, the indicated dilutions of the mistletoe extracts seem somewhat undefined, except that they were derived from a 1:20 dilution of the respective mother tincture. Is there a way to determine the concentration of the extracts or at least to assess the comparability of the different extracts? Otherwise, analog dilutions can only be compared if the initial concentration was the same.

Furthermore, the question arises why the authors did not present the cell viability determined using the MTT assay as such, but instead converted it into cell numbers. This may be associated with uncertainties and may not even be necessary, as proliferation was also measured in parallel using the BrdU assay.

The results are presented in a clearly structured and consecutive manner. However, in the bar charts in Figure 5 (which unfortunately have poor resolution), the values should be shown with increasing concentration (i.e. decreasing dilution). If the apoptosis measurements were carried out in multiple determination, why is no statistical evaluation of the mean values shown, but only a representative example? When listing the changes in the CD44 variants and the integrins in the various cell lines, some things do not seem correct and should therefore be checked again (i.a. “β3 was particularly strong on PC3” or “However, CD44v4 was elevated in this cell line by Populi.” or “CD44v3, v5, and v6 were down-regulated by both Populi and Salicis, whereas only Populi reduced CD44v4 and v7.” in LNCaP cells!).

The authors list some possible reasons for the varying effects of the different mistletoe extracts in the different PCa cell lines in the discussion chapter. Subsequently, they discuss some potential active substances contained in the extracts which might be responsible for the anti-tumor action in vitro. However, it would be more interesting to know how much of these substances are contained in the various extracts and what differences there are in the composition of the analyzed extracts. Is it known what specific influence the host tree can have on the mistletoe? The authors then discuss other cellular factors that could be influenced by mistletoe extracts, such as Akt and PTEN. Wasn't it possible to analyze these signaling pathways in the study setting to gain clarity about their involvement? Since the authors propose that mistletoe extracts inhibit PCa growth through integrin α5 and α6 suppression, a further and more in-depth analysis of these factors should follow.

Author Response

Answers to the comments of referee 1

Comment 1: Why they chose PCa as a tumor model should be better explained, as there seems to be no evidence of the efficacy of mistletoe preparations for this tumor entity so far. In addition, the clinical situation in which such preparations could be used in PCa should be considered more thoroughly.

Our answer: Mistletoe is quite popular among cancer patients, particularly in German speaking countries. However, the therapeutic relevance for treating cancer patients is still debatable. The S3-guideline for complementary medicine in treating oncological patients points to discrepancies. Some clinical trials report superior overall survival with mistletoe, whereas others do not (https://www.leitlinienprogramm-onkologie.de/leitlinien/komplementaermedizin). Studies have been done with different tumor entities (e.g. Paller et al. Phase I Trial of Intravenous Mistletoe Extract in Advanced Cancer. doi: 10.1158/2767-9764.CRC-23-0002. Cogo et al. Mistletoe Extracts during the Oncological Perioperative Period: A Systematic Review and Meta-Analysis of Human Randomized Controlled Trials. doi: 10.3390/curroncol30090595.), however, data with regard to prostate cancer are sparse. This is surprising, since mistletoe is used by nearly 10% of prostate cancer patients (related to CAM users in Austria; Ponholzer et al. Frequent use of complementary medicine by prostate cancer patients. doi: 10.1016/s0302-2838(03)00155-6.). Büssing et al. assumed that mistletoe extracts may provide benefit in prostate cancer patients when given on an individual basis (Büssing et al. Course of mitogen-stimulated T lymphocytes in cancer patients treated with Viscum album extracts. Anticancer Res. 2007 27(4C):2903-10.). There is evidence that mistletoe may cause an additive inhibitory effect on prostate cancer cells when combined with chemotherapy (Weissenstein et al. Interaction of standardized mistletoe (Viscum album) extracts with chemotherapeutic drugs regarding cytostatic and cytotoxic effects in vitro. doi: 10.1186/1472-6882-14-6.).

The discovery that mistletoe reinforces the immune system has now opened novel and exciting treatment options (Hong et al. Mistletoe in Cancer Cell Biology: Recent Advances. doi: 10.3390/cimb47080672. Nicoletti M. The Anti-Inflammatory Activity of Viscum album. doi: 10.3390/plants12071460.). Indeed, two phase I studies have been initiated, documenting safety of mistletoe application in patients with advanced cancer (including prostate cancer patients; Huber et al. Safety of intravenously applied mistletoe extract - results from a phase I dose escalation study in patients with advanced cancer. doi: 10.1186/s12906-017-1971-1.), along with stable disease and tumor shrinkage in some heavily pretreated patients. (Paller et al. Phase I Trial of Intravenous Mistletoe Extract in Advanced Cancer. doi: 10.1158/2767-9764.CRC-23-0002.). However, the authors pointed to the “preliminary” nature of their data and recommended further investigation.

To address the referee’s comment we have added to the introduction (lines 83-92): “Whether mistletoe can be used to treat PCa remains to be explored. In fact, although studies have been done with different tumor entities [28,29], investigation with regard to PCa is sparse. This scarcity is not justified since mistletoe is applied by nearly 10% of PCa patients (related to CAM users in Austria) [30]. Büssing et al. assumed that mistletoe extracts may provide benefit in PCa patients when given on an individual basis [31]. Evidence has shown that mistletoe may cause an additive inhibitory effect on PCa cells when combined with chemotherapy [32].

Two phase I studies have meanwhile been initiated, documenting safety of mistletoe application in PCa patients [33] along with stable disease and tumor shrinkage in some heavily pretreated patients [28]. However, the authors pointed to the “preliminary” nature of their data and recommended further investigation [28]. Therefore, we initiated a comparative in vitro study with a panel of PCa cell lines to explore the growth and proliferation blocking potential of mistletoe preparations derived from four different host trees, Tiliae, Populi, Salicis, and Crataegi”.

Comment 2. If there are further reports than those already mentioned on which cellular processes mistletoe extracts may have an influence, these should be described in the introduction. There may also be more references on the role of CD44 and integrins in PCa already in the introduction. This would better justify the more detailed consideration of these and no other molecules and pathways.

Our answer: We have added information on the molecular mode of action of mistletoe, and added some further references highlighting the role of CD44 and integrins in prostate cancer. “Introduction” now reads (lines 78-80 and lines 98-101): “Similar effects have been observed with further solid tumor types including liver [23], gastric [24], and lung cancer cells [25]. Mistletoe has been proven to target the mitogen-activated protein kinase (MAPK) and the PI3K/Akt pathway related to growth and survival of the cancer cells [26]. Furthermore, mistletoe extracts inhibited bladder cancer cell growth and proliferation by acting on the cyclin-CDK axis and the expression level of membrane receptors associated with tumor cell differentiation [27]”.

“Besides assessing growth and proliferation, cell cycle progression and the expression of proteins involved in cell cycle regulation (cyclin-dependent kinases, cyclins) were analyzed in mistletoe treated versus non-treated PCa cells. This was done by investigating integrin α and β subtypes that have been shown to regulate PCa growth, progression, and metastatic formation [34]. Since the CD44s and CD44v isoforms are also involved in PCa proliferation, invasion, and metastasis [35], they were evaluated as well.

Comment 3: The methods are adequately introduced and explained. However, the indicated dilutions of the mistletoe extracts seem somewhat undefined, except that they were derived from a 1:20 dilution of the respective mother tincture. Is there a way to determine the concentration of the extracts or at least to assess the comparability of the different extracts? Otherwise, analog dilutions can only be compared if the initial concentration was the same.

Our answer: The mistletoe extracts were prepared according to an established protocol (please see reference 36). We have now added a further reference documenting the preparation process (Jennifer et al. Investigations of Willow Mistletoe on Various Human Tumor Cell Lines in vitro. doi: 10.1055/a-0799-7042). “Materials and Methods”, chapter “Mistletoe extracts”, now reads (lines 115-119): “The concentrations of mistletoe lectin and viscotoxin were as follows: lectin-Tiliae: 338.54 ng/ml, lectin-Populi: 184.33 ng/ml, lectin-Salicis: 246.46 ng/ml, lectin-Crataegi: 211.49 ng/ml, viscotoxin-Tiliae: 1.046 µg/ml, viscotoxin-Populi: 878 µg/ml, viscotoxin-Salicis: 980 µg/ml, viscotoxin-Crataegi: 1.109 µg/ml. The extracts, prepared according to an established protocol [36,37], ”.

All dilutions used in the experiments were derived from the same batch. This was necessary to guarantee analog concentrations with the same extract composition. To make this point clear, we have now added (lines 124-125): “The employed dilutions were as follows, with (E) and (F) designating the dilutions used to evaluate proliferation (after 48 h incubation), or (E) to evaluate cell cycling (after 24 h incubation), apoptosis (after 48 h incubation), and integrin/CD44 expression (after 24 h incubation). To guarantee identical concentrations with the same extract composition, all dilutions were prepared from the same mistletoe batch”.

Comment 4: Furthermore, the question arises why the authors did not present the cell viability determined using the MTT assay as such, but instead converted it into cell numbers. This may be associated with uncertainties and may not even be necessary, as proliferation was also measured in parallel using the BrdU assay.

Our answer: It is correct that the MTT assay represents an employed technique to assess cell viability. However, the intensity of the blue formazan colour can be related to cell number as well (Khalef et al. Cell viability and cytotoxicity assays: Biochemical elements and cellular compartments. doi: 10.1002/cbf.4007.). Since we specifically investigated whether mistletoe may influence tumor growth, we have related the MTT data to tumor cell number. In this context, we would like to emphasize that apoptotic events have also been recorded including the presentation of viable cell alterations in the presence of the mistletoe compounds. We, therefore, would like to depict the MTT data as tumor cell number, although we are prepared to change to “viability” if the referee still insists upon this.

Since the MTT assay only allows conclusions as to tumor cell number but not to proliferation, we additionally did the BrdU assay which exclusively evaluates the process of DNA synthesis in actively dividing cells.

Comment 5: The results are presented in a clearly structured and consecutive manner. However, in the bar charts in Figure 5 (which unfortunately have poor resolution), the values should be shown with increasing concentration (i.e. decreasing dilution). If the apoptosis measurements were carried out in multiple determination, why is no statistical evaluation of the mean values shown, but only a representative example? When listing the changes in the CD44 variants and the integrins in the various cell lines, some things do not seem correct and should therefore be checked again (i.a. “β3 was particularly strong on PC3” or “However, CD44v4 was elevated in this cell line by Populi.” or “CD44v3, v5, and v6 were down-regulated by both Populi and Salicis, whereas only Populi reduced CD44v4 and v7.” in LNCaP cells!).

Our answer: Figure 5 now depicts the values in increasing concentrations. Apoptosis was measured several times to allow statistic evaluation. So as to not overload figure 5, we decided to include just the mean values. We have now provided the inter-assay variation in the figure legend which now reads (lines 328-329): “Figure shows one representative from n=4. Mean values are given as percentage, SDintraassay < 30%.” CD44 and integrin data have been checked and corrected. We apologize for these mistakes. The respective results section now reads (lines 346-347): “…..β3 was particularly strongly expressed on DU145”. (lines 358-360) “However, CD44v5 was elevated in this cell line by Populi”. “CD44v3, v5, and v6 were down-regulated by both Populi and Salicis, whereas only Populi reduced CD44v4 and v7 in LNCaP cells (Fig.8C)”.

Comment 6: The authors list some possible reasons for the varying effects of the different mistletoe extracts in the different PCa cell lines in the discussion chapter. Subsequently, they discuss some potential active substances contained in the extracts which might be responsible for the anti-tumor action in vitro. However, it would be more interesting to know how much of these substances are contained in the various extracts and what differences there are in the composition of the analyzed extracts. Is it known what specific influence the host tree can have on the mistletoe? The authors then discuss other cellular factors that could be influenced by mistletoe extracts, such as Akt and PTEN. Wasn't it possible to analyze these signaling pathways in the study setting to gain clarity about their involvement? Since the authors propose that mistletoe extracts inhibit PCa growth through integrin α5 and α6 suppression, a further and more in-depth analysis of these factors should follow.

Our answer: We have now added more information about the mistletoe lectin and viscotoxin contents of the compounds used in this study. “Materials and Methods”, chapter “Mistletoe extracts”, now reads: (lines 115-119) “ ….. in an isotonic solution. The concentrations of mistletoe lectin and viscotoxin were as follows: lectin-Tiliae: 338.54 ng/ml, lectin-Populi: 184.33 ng/ml, lectin-Salicis: 246.46 ng/ml, lectin-Crataegi: 211.49 ng/ml, viscotoxin-Tiliae: 1.046 µg/ml, viscotoxin-Populi: 878 µg/ml, viscotoxin-Salicis: 980 µg/ml, viscotoxin-Crataegi: 1.109 µg/ml.

We have also added to the Discussion (lines 438-442) : “In particular, the content of mistletoe lectin and viscotoxin varies among the host tree species and between summer and winter harvesting [49-50]. Both compounds are probably related to the antitumor effect of mistletoe extracts. Viscotoxin and lectin concentrations also differed in the mistletoe extracts used in the present study. However, viscotoxin was highest in extracts derived from Crataegi, whereas lectin was highest in extracts derived from Tiliae, making it difficult to conclude to a relationship between these substances and tumor cell response. In this context, it has been postulated by others that the whole mistletoe content in concert, rather than single compounds, exerts effects on tumor cells. Analysis of …. may not be exclusively related to the viscotoxin and lectin content but to the composition of the total phytocomplex [51].”

The referee is correct that further signaling pathways may also be influenced by mistletoe, e.g. the Akt-mTOR and MAPK route (Szurpnicka et al. Biological activity of mistletoe: in vitro and in vivo studies and mechanisms of action. doi: 10.1007/s12272-020-01247-w.). We have addressed this point in the discussion section. In fact, we have already documented that integrins may be linked to Akt/mTOR (Ref. 39). Concerning the present paper, we believe that additional investigation of Akt/mTOR (including rictor and raptor) expression and activation would considerably overload the present report. However, it makes sense to evaluate in more depth whether integrins α5 and α6 are connected to Akt/mTOR. We have added to the Discussion (lines 532-533): “Based on the in vitro data in the present investigation, we propose that mistletoe extracts inhibit PCa growth through integrin α5 and α6 suppression. In-depth analyses now follow to explore whether integrins α5 and α6 are connected to the Akt/mTOR pathway”.

Reviewer 2 Report

Comments and Suggestions for Authors

Journal: Cells
Manuscript ID: cells-3793932
Type of manuscript: Article

Title: Mistletoe extracts inhibit progressive growth of prostate cancer cells

Authors: Sascha D. Markowitsch, Larissa Albrecht, Moritz Meiborg, Jochen Rutz, Anita Thomas, Felix K.-H. Chun, Axel Haferkamp, Eva Juengel, Roman A. Blaheta

The present investigation was designed to evaluate the effects of mistletoe extracts from four different host trees, Tiliae, Populi, Salicis, and Crataegi, on the growth and proliferation of PC3, DU145, and LNCaP human prostate cancer cells. Clonogenicity, apoptosis, cell cycle, and cell cycle regulating proteins were also investigated. The results showed that all the mistletoe extracts inhibited significantly the prostate cancer cell growth and proliferation, while Populi and Salicis extracts induced cell cycle arrest in the G2/M phase and increased apoptosis, and they down-regulated CDK1 and cyclin A and altered CD44 expression. These indicate that these mistletoe extracts could be used potentially as an adjunct to standard prostate treatment. This article could be published as a potential Article in Cells after a few minor changes. In the introduction section, antitumor potentials of the Tiliae, Populi, Salicis, and Crataegi extracts reported in literature should be summarized, and the research work related to the present study could be highlighted. The authors showed that all the four mistletoe extracts exhibited activity against three human prostate cancer cell lines tested (Figures 1 and 2), but they only selected Populi and Salicis extracts for further biological investigations (Figures 3‒5 and 8). The reasons for this selection should be discussed briefly in the manuscript.

Author Response

Answers to the comments of referee 2

Comment 1: In the introduction section, antitumor potentials of the Tiliae, Populi, Salicis, and Crataegi extracts reported in literature should be summarized, and the research work related to the present study could be highlighted.

Our answer: We have now added summarized information on the molecular mode of action of mistletoe. “Introduction” now reads (lines78-80): “Similar effects have been observed with further solid tumor types including liver [23], gastric [24], and lung cancer cells [25]. Mistletoe has been proven to target the mitogen-activated protein kinase (MAPK) and the PI3K/Akt pathway related to growth and survival of the cancer cells [26]. Furthermore, mistletoe extracts inhibit bladder cancer cell growth and proliferation, presumably by acting on the cyclin-CDK axis and the expression level of membrane receptors associated with tumor cell differentiation [27]”.

We also better explain our rationale concentrating on prostate cancer cells, and have now added to the Introduction (lines 83-92) : “Whether mistletoe can be used to treat PCa remains to be explored. In fact, although studies have been done with different tumor entities [28,29], investigation with regard to PCa is sparse. This scarcity is not justified since mistletoe is applied by nearly 10% of PCa patients (related to CAM users in Austria) [30]. Büssing et al. assumed that mistletoe extracts may provide benefit in PCa patients when given on an individual basis [31]. Evidence has shown that mistletoe may cause an additive inhibitory effect on PCa cells when combined with chemotherapy [32].

Two phase I studies have meanwhile been initiated, documenting safety of mistletoe application in PCa patients [33] along with stable disease and tumor shrinkage in some heavily pretreated patients [28]. However, the authors pointed to the “preliminary” nature of their data and recommended further investigation [28]. Therefore, we initiated a comparative in vitro study with a panel of PCa cell lines to explore the growth and proliferation blocking potential of mistletoe preparations derived from four different host trees, Tiliae, Populi, Salicis, and Crataegi”.

Comment 2: The authors showed that all the four mistletoe extracts exhibited activity against three human prostate cancer cell lines tested (Figures 1 and 2), but they only selected Populi and Salicis extracts for further biological investigations (Figures 3‒5 and 8). The reasons for this selection should be discussed briefly in the manuscript.

Our answer: Overall, we did not detect strong differences among the extracts used. However, Populi and Salicis caused distinct effects in the BrdU proliferation assay on DU145 cells. Therefore, we decided to concentrate on these extracts in the ongoing experiments. This strategy is explained in the results section, 3.1 Extracts from mistletoe inhibit tumor cell growth and proliferation of PCa cells (lines 273-276): “Particularly, Populi completely stopped BrdU uptake in DU145 when added at dilution F. A very strong response was induced when DU145 cells were exposed to Salicis. Therefore, ongoing studies concentrated on Populi and Salicis”.

Round 2

Reviewer 1 Report

Comments and Suggestions for Authors

The reviewer's concerns were addressed accordingly.